# Unbiased Stochastic Optimization for Gaussian Processes on Finite Dimensional RKHS

## Abstract

Current methods for stochastic hyperparameter learning in *Gaussian Processes* (GPs) rely on approximations, such as computing biased stochastic gradients or using inducing points in stochastic variational inference. However, when using such methods we are not guaranteed to converge to a stationary point of the true marginal likelihood. In this work, we propose algorithms for exact stochastic inference of GPs with kernels that induce a Reproducing Kernel Hilbert Space (RKHS) of moderate finite dimension. Our approach can also be extended to infinite dimensional RKHSs at the cost of forgoing exactness. Both for finite and infinite dimensional RKHSs, our method achieves better experimental results than existing methods when memory resources limit the feasible batch size and the possible number of inducing points.

## 1 Introduction

Gaussian Processes (GPs) provide a powerful probabilistic framework which has been applied to a wide range of learning applications, such as multi-task learning (Alvarez et al., 2012; Liu et al., 2018a), active learning (Liu et al., 2018b), semi-supervised learning Jean et al. (2018), and reinforcement learning (Srinivas et al., 2010; Shahriari et al., 2015). These successes can be attributed to the natural way in which uncertainty is incorporated into the predictions via a Bayesian interpretation. However, hyperparameter learning scales poorly. For a general covariance function (kernel function in kernel method's parlance) the computational cost grows as $O(n^3)$ where $n$ denotes the number of samples, and the storage resources grow as $O(n^2)$. As a consequence, approximations are required for any modern application that involves GPs and big data.

It is therefore unsurprising that much research effort has been devoted to approximate methods for learning GPs. For a comprehensive overview of this research, we refer the reader to the work of Liu et al. (2020). The most expensive operation in training GPs is the inversion of the $n \times n$ kernel matrix which is required during the maximization of the marginal likelihood and for the computation of the posterior of the responses. As a result, much research has been devoted to providing a cheap approximation of the kernel matrix.

Broadly speaking approximations of GPs can be split in to *data dependent* and *data independent* methods. Data dependent approximations usually come with valuable probabilistic interpretations. An important work in this line of research is the seminal work of Quinonero-Candela & Rasmussen (2005), which provides a unified view on previous work by using the concept of inducing points. Since then quite a few followup works have used inducing points, while exploiting the tool of variational inference as a theoretical platform (Titsias, 2009; Nguyen et al., 2014; Wilson & Nickisch, 2015; Zhao & Sun, 2016). This approach is closely related to the Nyström approximation of kernel matrices (Zhao & Sun, 2016).

In contrast, data independent methods rely on approximating the kernel function itself. Typically, it is approximated by an inner product between low dimensional vectors (Rahimi & Recht, 2007; Yang et al., 2015; Shustin & Avron, 2021).

Approximating the inverse of the kernel matrix is enough for frequentist kernel methods such as Kernel Ridge Regression. However, in order to harness the full power of Bayesian kernel methods such as *Gaussian Process Regression* (GPR), we need to be also able to maximize the marginal likelihood, and for that an

approximation of the inverse of the kernel matrix is not enough, due to an additional log-determinant term. This is especially true in cases where the covariance function depends on a large number of parameters, e.g., evaluating the covariance function involves a forward pass of a deep neural network (Wilson et al., 2016b; Calandra et al., 2016; Wilson et al., 2016a). In such cases, it is important to be able to use a stochastic mini-batches based optimization, since otherwise the cost of making a pass over the entire dataset typically proves too expensive. Moreover, it is well appreciated in the literature that when the model involves deep neural networks, e.g., when the covariance function is defined by a neural network, it is important to use stochastic gradients, since they enable more efficient optimization and better generalization (Goodfellow et al., 2016).

Currently, two main approaches are used for stochastic hyperparameter learning in GPs. The first relies on the inducing-points framework and employs *stochastic variational inference* (SVI) (Hoffman et al., 2013; Hensman et al., 2013; Hoang et al., 2015; Wilson et al., 2016a). The second approach is more direct: it is based on computing stochastic batches while ignoring the fact that they provide only biased estimates of gradients. Interestingly, a recent work shows that despite the bias, given a large enough batch size, the direct approach produces almost optimal models (in terms of the marginal likelihood) (Chen et al., 2020).

Both the SVI approach and the direct approach suffer from several disadvantages. For example, consider the case in which the covariance function is the inner product between two feature maps of a moderate dimension. This can be either because this decomposition serves as an approximation to another covariance function with an infinite dimensional *reproducing kernel Hilbert space* (RKHS), or because it has been defined this way to begin with, e.g., as an inner product between features that are created by passing the data through a neural network. Either way, using the SVI approach amounts to imposing an additional unnecessary approximation that comes from the need to use inducing points. As for the second approach, its applicability is highly dependent on our ability to use large batches. This can be a serious impediment in several cases, such as optimization on weak edge devices (Chen et al., 2016).

In this paper, we propose two stochastic optimization algorithms based on mini-batches for maximizing the marginal likelihood of GPs (i.e., learning the hyperparameters). The first algorithm is based on reframing the problem as a nonconvex-concave minimax problem. We then leverage recent advancements in the theory of solving such problems (Boţ & Böhm, 2020; Lin et al., 2020; Luo et al., 2020) to propose a concrete algorithm. The second algorithm is based on writing the loss function in compositional form, i.e., as the composition of a function and the expected value of another function. We then use the recently introduced Stochastic Compositional Gradient Descent method (Wang et al., 2017). Our novel algorithms are applicable for covariance functions connected with an RKHS of a moderate dimension, and guarantee convergence of the marginal likelihood to a local minimum for any batch size without need for further approximations. In the infinite dimensional case (e.g., Gaussian covariance function) one can use our method on top of a low rank approximation of the covariance function, e.g., using the random features method Rahimi & Recht (2007). Our experiments show that not only is our method superior to existing methods for stochastic optimization of the marginal likelihood in the finite dimensional case when the batches have a moderated size, it is also superior to the existing methods in the infinite dimensional case if the restriction on the batch size is more severe, although in the infinite dimensional case the models found by our method are no longer optimal.

**Additional Related Work** Apart from the extensive literature on scaling GPs using approximations, several works are focused on exact inference using sophisticated distributed algorithms (Nguyen et al., 2019; Wang et al., 2019). In this context, the biased stochastic gradient proposed by Chen et al. (2020) can be considered as an economical method for exact inference, given that the covariance function and the system enable computation in large enough batches.

## 2 Preliminaries

### 2.1 Notations and Basic Definitions

For a function $f : \mathcal{U} \to \mathbb{R}$ and $U = (u_1, \ldots, u_m) \in \mathcal{U}^m$, $\mathbf{f} = f(U)$ is a vector in $\mathbb{R}^m$ such that $\mathbf{f}_i = f(u_i)$. Similarly, for a binary function $k : \mathcal{U} \times \mathcal{U} \to \mathbb{R}$, $K = k(U, U)$ is a matrix in $\mathbb{R}^{m \times m}$ such that $K_{ij} = k(u_i, u_j)$.

Given a size $m$ vector $\mathbf{b}$ and $\mathcal{S} = (s_1, \ldots, s_p) \in \{1, \ldots, m\}^p$, we use $\mathbf{b}_\mathcal{S}$ to denote the size $p$ vector such that the $i$'th coordinate of $\mathbf{b}_\mathcal{S}$ is equal to the $s_i$'th coordinate of $\mathbf{b}$. In a similar way if $C$ is an $m \times n$ matrix then $C_\mathcal{S}$ is a $p \times n$ matrix such that the $i$'th row of $C_\mathcal{S}$ is equal to the $s_i$'th row of $C$. Finally, if $\mathcal{R} = (r_1, \ldots, r_q) \in \{1, \ldots, n\}^q$ then $C_{\mathcal{S}\mathcal{R}}$ is a $p \times q$ matrix such that the $(i, j)$ coordinate of $C_{\mathcal{S}\mathcal{R}}$ is equal to the $(s_i, r_j)$ coordinate of $C$.

For a square matrix $A$ we use the notation $|A|$ to denote the determinant of $A$. If $\mathcal{S}$ is a finite sequence or a set then $|\mathcal{S}|$ denotes its length or size. Whenever we use $\langle A, B \rangle$ where $A$ and $B$ are real matrices, $\langle \cdot, \cdot \rangle$ symbolizes the Frobenius inner product which is defined as

$$\langle A, B \rangle = \mathbf{Tr}\left(AB^T\right).$$

If $A$ is a real matrix then $\|A\|$ is the Frobenius norm of $A$ so,

$$\|A\| = \sqrt{\langle A, A \rangle}.$$

For a vector $\mathbf{v}$, $\|\mathbf{v}\|$ is the Euclidean norm of $\mathbf{v} \in \mathbb{R}^q$.

For any closed convex set $\Omega \subseteq \mathbb{R}^q$ and $\mathbf{v} \in \mathbb{R}^q$, $\mathrm{proj}_\Omega(\mathbf{v}) = \arg\min_{\mathbf{v}' \in \Omega} \|v' - v\|$ is the Euclidean projection of $\mathbf{v}$ on $\Omega$.

## 2.2 Hyperparameter Learning in Gaussian Process Regression

Let $\mathcal{X}$ be a feature space and let $k_\alpha : \mathcal{X} \times \mathcal{X} \to \mathbb{R}$ be a positive definite covariance function parameterized by hyperparameters $\alpha \in \mathbb{R}^m$. For any $\alpha \in \mathbb{R}^m$, let $f_\alpha$ be a random function on $\mathcal{X}$ distributed as a zero mean GP whose covariance is $k_\alpha$, that is, for any $j \in \mathbb{N}$, $U \in \mathcal{X}^j$:

$$f_\alpha(U) \sim \mathcal{N}\left(0, k_\alpha(U, U)\right).$$

Let $X \in \mathcal{X}^n$, $\mathbf{y} \in \mathbb{R}^n$ be a training set, where $n$ is the number of training samples. In *Gaussian Process Regression* (GPR), it is assumed that $\mathbf{y}$ is a sample of the random vector $f_{\bar{\alpha}}(X) + \bar{\sigma}^2 \epsilon$ where $\epsilon \sim \mathcal{N}(0, I_n)$, and $\bar{\alpha}, \bar{\sigma}^2$ are the true hyperparameters of the model.

In GPR, hyperparameters are learned by solving a maximum likelihood type II problem, i.e., maximizing the marginal likelihood.

$$p(\mathbf{y}|X, \alpha, \sigma) = \mathcal{N}\left(\mathbf{y}|0, K(\alpha) + \sigma^2 I\right)$$

with respect to $\alpha$ and $\sigma$, where $K(\alpha) = k_\alpha(X, X)$. The maximization of the marginal likelihood $p(\mathbf{y}|X, \alpha, \sigma)$ is equivalent to the minimization of

$$\mathbf{y}^T \left(K(\alpha) + \sigma^2 I\right)^{-1} \mathbf{y} + \log\left|K(\alpha) + \sigma^2 I\right|.$$

See Rasmussen (2003) for details. In this work we further assume that the covariance function has the following form:

$$k_\alpha(x, x') = \phi_\alpha(x)^T \phi_\alpha(x')$$

for some feature map $\phi_\alpha : \mathcal{X} \to \mathbb{R}^d$. It can be shown that for all $\lambda > 0$, $V \in \mathbb{R}^{n \times d}$ and $\mathbf{b} \in \mathbb{R}^d$ we have that

$$\mathbf{b}^T \left(VV^T + \lambda I\right)^{-1} \mathbf{b} = \min_{\mathbf{w}} \frac{1}{\lambda} \|V\mathbf{w} - \mathbf{b}\| + \|\mathbf{w}\|^2$$

(see Appendix B). As a result, maximizing $p(\mathbf{y}|X, \alpha, \sigma)$ is equivalent to minimizing

$$l(\theta) = \frac{1}{\sigma^2} \|Z(\alpha)\mathbf{w} - \mathbf{y}\|^2 + \|\mathbf{w}\|^2 + \log|F(\theta)| + (n - d)\log\sigma^2,$$

where

$$\theta = \left(\mathbf{w}, \alpha, \sigma^2\right),$$
$$Z(\alpha) = \left(\phi_\alpha(x_1), \ldots, \phi_\alpha(x_n)\right)^T \in \mathbb{R}^{n \times d},$$
$$F(\theta) = Z(\alpha)^T Z(\alpha) + \sigma^2 I \in \mathbb{R}^{d \times d}.$$

The goal of this work is to propose algorithms for minimizing $l(\theta)$ using a stochastic gradient method based on mini-batches. This would have been straightforward if we could write

$$\nabla_\theta \log |F(\theta)| = \sum_{i=1}^{n} G(\theta; x_i)$$

for some function $G(\cdot; \cdot)$. However, there is no obvious decomposition of this form.

## 3 Stochastic Optimization for Gaussian Processes

This section contains our main contribution: two novel stochastic mini-batched based algorithms for minimizing $l(\theta)$. Before detailing our approaches, we make few additional notations:

$$g_i(\theta) = \frac{1}{\sigma^2} (\phi_\alpha(x_i)\mathbf{w} - y_i)^2 + \frac{1}{n} \|\mathbf{w}\|^2 + \frac{1}{n} (n - d) \log \sigma^2$$

$$g(\theta) = \sum_{i=1}^{n} g_i(\theta)$$

$$F_i(\theta) = \phi_\alpha(x_i) \phi_\alpha(x_i)^T + \frac{1}{n} \sigma^2 I$$

$$h(A) = \log |A|$$

So we can write,

$$F(\theta) = \sum_{i=1}^{n} F_i(\theta)$$

$$l(\theta) = g(\theta) + h(F(\theta)).$$

### 3.1 A Minimax Approach

Our first step is to replace the problem

$$\min_\theta g(\theta) + h(F(\theta))$$

with the equivalent problem

$$\min_{\theta, A} g(\theta) + h(A)$$

$$\text{s.t. } A = F(\theta) \tag{1}$$

The next step is to replace the last problem with a parameterized problem, such that the hard constraint is replaced with a penalty term, and the penalty term is driven to infinity. To do so, let us first define the optimization problems

$$\min_\zeta l_\mu(\zeta)$$

where $\zeta = (\theta, A)$ and

$$l_\mu(\zeta) = g(\theta) + h(A) + \mu \frac{\|A - F(\theta)\|}{\|A\|}.$$

Now, suppose that Problem (1) admits an optimal solution. Let the sequences $\{\mu_k\}$ and $\{\zeta_k\}$ satisfy $\mu_k \to \infty$ and, for each $k$, let $\zeta_k$ minimize $l_{\mu_k}(\zeta)$. If $\zeta^\star$ is an accumulation point of $\{\zeta_k\}$, then $\zeta^\star$ is an optimal solution of Problem (1) (Ruszczynski, 2006, Theorem 6.6).

The reason we use

$$\frac{\|A - F(\theta)\|}{\|A\|}$$

instead of just $\|A - F(\theta)\|$ is to avoid finding critical points around $F(\theta) = 0$.

Unfortunately, the term $\|A - F(\theta)\|$ exhibits the same issues as $h(F(\theta)) = \log \det F(\theta)$ and does not allow straightforward unbiased mini-batch based stochastic gradients.

In order to handle this, we use the fact that for an element $v$ in Euclidean space we have that $\|v\| = \max_{\|u\| \le 1} \langle u, v \rangle$. Thus, minimizing $l_\mu(\zeta)$ is equivalent to

$$\min_\zeta \max_{\|B\| \le 1} \Psi(\zeta, B), \tag{2}$$

where

$$\Psi(\zeta, B) = g(\theta) + h(A) + \mu \frac{\langle B, A - F(\theta) \rangle}{\|A\|}$$

Now, we can write

$$\Psi(\zeta, B) = \sum_{i=1}^n \psi(\zeta, B; x_i) \tag{3}$$

where

$$\psi(\zeta, B; x_i, \mu) = g_i(\theta) + \frac{1}{n} h(A) + \mu \frac{\langle B, \frac{1}{n} A - F_i(\theta) \rangle}{\|A\|}$$

Thus, Problem (2) naturally admits unbiased stochastic gradients based on mini-batches taken separately for the minimization and the maximization parts.

For better stability, we restrict $A$ to be positive semidefinite such that $A \succeq \sigma^2 I$, and restrict $\sigma \ge \sigma_{\min}$, where $\sigma_m in$ is a non-learned hyperparameter. To further restrict the search space, we constrain $\zeta \preceq \zeta_{\max}$, where $\zeta_{\max}$ is another non-learned hyperparameter. Together, the constraints on $\zeta$ can be represented by $\zeta \in \Omega_1$, where $\Omega_1$ is a convex set.

Let $\Omega_2 = \{ B \in \mathbb{R}^{d \times d} \mid \|B\| \le 1 \}$. The algorithm we propose is based on the following update rule for solving Problem (2):

$$1. \ \zeta_{t+1} = \text{proj}_{\Omega_1} \left( \zeta_k - a \nabla_\zeta \frac{n}{s} \sum_{i \in \mathcal{S}_{t+1}} \psi(\zeta_t, B_t; x_i, \mu_t) \right)$$

$$2. \ B_{t+1} = \text{proj}_{\Omega_2} \left( B_t + b \nabla_B \frac{n}{s} \sum_{i \in \bar{\mathcal{S}}_{t+1}} \psi(\zeta_{t+1}, B_t; x_i, \mu_t) \right)$$

where $\mathcal{S}_t$ and $\bar{\mathcal{S}}_t$ are sets of $s$ indices randomly chosen from $\{1, \ldots, n\}$ independently from all previous iterations. Recent work by Boţ & Böhm (2020) shows that in case of a minimax problem where maximization is of a concave function over a convex constraint[1], then with a few additional mild conditions on the objective function, an algorithm based on the above update rule converges to a stationary point in $O(\epsilon^{-8})$ iterations.

### 3.2 Stochastic Compositional Gradient Descent Approach

Consider a loss function $l : \Theta \to \mathbb{R}$ of the form $l(\theta) = v(u(\theta))$ where $u : \Theta \to \mathbb{R}^p$ and $v : \mathbb{R}^p \to \mathbb{R}$ are differentiable functions, and assume that $u(\theta) = \mathbb{E}_\omega \tilde{u}(\theta; \omega)$ for a differentiable function $\tilde{u}(\theta; \omega)$ that depends on a random variable $\omega$. *Stochastic Compositional Gradient Descent* (SCGD) (Wang et al., 2017) is an intuitive algorithm that alternates between two steps: updating the solution $\theta_t$ by a stochastic gradient iteration, and estimating $u(\theta_t)$ using an iterative weighted average of past values. More precisely, the update rule of of SCGD is given by:

$$1. \ \theta_{t+1} = \theta_t - a_t \langle \nabla v(\eta_t), \nabla \tilde{u}(\theta; \omega_t) \rangle$$
$$2. \ \eta_{t+1} = (1 - b_t) \eta_t + b_t \tilde{u}(\theta; \omega_t).$$

---

[1]The setup in Boţ & Böhm (2020) is more general.

where $\omega_1, \omega_2, \ldots$ are samples from $\omega$ in an i.i.d. manner, and $a_0, a_1, \ldots, b_0, b_1, \ldots$ degrees of freedom in the algorithm. Under few additional standard conditions on $u, v, \tilde{u}$, Wang et al. (2017) showed a convergence rate of $\mathcal{O}\left(\epsilon^{-4}\right)$ if we choose $a_t = t^{-\frac{3}{4}}$ and $b_t = t^{-\frac{1}{2}}$, see (Wang et al., 2017, Theorem 8).

In order to use SCGD for our loss, we define:

$$u\left(\theta\right) = \left(g\left(\theta\right), F\left(\theta\right)\right),$$
$$v\left(u_1, u_2\right) = u_1 + h\left(u_2\right),$$
$$\tilde{u}\left(\theta; \omega\right) = \frac{n}{|\mathcal{S}|} \sum_{i \in \mathcal{S}} \left(g_i\left(\theta\right), F_i\left(\theta\right)\right)$$

where $\omega = \mathcal{S}$ is a random set of indices chosen from $\{1, \ldots, n\}$. With that, given the fact that

$$\frac{\partial \log |A|}{\partial A} = A^{-1}$$

we can write an explicit update rule for our SCGD-based algorithm:

1. $\theta_{t+1} = \theta_t - a_t \sum_{i \in \mathcal{S}} \nabla \left[g_i\left(\theta_t\right) + \left\langle \tilde{F}_t^{-1}, F_i\left(\theta\right)\right\rangle\right]$

2. $\tilde{F}_{t+1} = \left(1 - b_t\right) \tilde{F}_t + b_t \frac{n}{|\mathcal{S}|} \sum_{i \in \mathcal{S}} F_i\left(\theta_t\right).$

## 4 Comparison to Existing Methods

In this work we suggest two novel methods for stochastic optimization of the marginal likelihood. We recognize two main competing methods that also optimize the marginal likelihood stochastically. The simplest among them is what we henceforth refer to as *Biased Stochastic Gradient Decent* (BSGD) (Chen et al., 2020). The idea in BSGD is to take the gradient of the marginal likelihood using data only from the current batch, ignoring the fact that this produces only a biased estimate of the full gradient. On the other hand *Scalable Variational Gaussian Process* (SVGP) of Hensman et al. (2015) is a sophisticated approach that approximates the original inference problem using inducing points and stochastic variational inference.

Unlike our algorithms, how well each of the competing methods approximates the solution of the true problem depends on memory consumption. In the case of BSGD, the bias in stochastic gradients shrinks as batch size increases. On the other hand, inference quality SVGP crucially depends on the number of inducing points which need to be processed forward and backward by a neural network at each iteration.

### 4.1 Complexity Analysis

Let C and M be the computational and the memory complexity in the computation of $\frac{\partial \phi_\alpha(x)}{\partial \alpha}$, and let $b$ be the batch size. For SVGP, assume that the number of the inducing points is in the same order of $b$.

**Number of inducing points vs. batch size.** For the sake if comparison, We choose the number of inducing points to be roughly the same as the mini-batch size because each inducing point requires a forward and backward pass through the network, storing activations and gradients just like a data sample. Thus, both sets consume memory in a similar way during each iteration, and matching their sizes keeps the per-iteration memory cost balanced and comparable.

The computational complexity of computing one optimization iteration of both BSGD and SVGP is $O\left(b^3\right) + O\left(b^2 d\right) + bC$ while both our algorithm take $O\left(d^3\right) + O\left(bd^2\right) + bC$ per iteration. For storage, BSGD and SVGP take $O\left(b^2\right) + O\left(bd\right) + bM$ while ours takes $O\left(d^2\right) + O\left(bd\right) + bM$.

Typically M is relatively large as it is the storage used by AD to compute the gradient of a neural network. In small devices that means one might have to use small batches. Our approach aims not to harm the exactness of the algorithm in this case. Naturally, to use our algorithms effectively, the feature dimension $d = \dim \phi_\alpha(x)$ must remain moderate, since memory scales as $O(d^2)$ and computation as $O(d^3)$.

Table 1: Complexity analysis for one optimization iteration: C and M are the computational and the memory complexity in the computation of $\frac{\partial \phi_\alpha(x)}{\partial \alpha}$. $b$ is the batch size.

| Algorithm | Computations | Storage |
|-----------|--------------|---------|
| SVGP | $O\left(b^3\right) + O\left(b^2 d\right) + b\text{C}$ | $O\left(b^2\right) + O\left(bd\right) + b\text{M}$ |
| BSGD | $O\left(b^3\right) + O\left(b^2 d\right) + b\text{C}$ | $O\left(b^2\right) + O\left(bd\right) + b\text{M}$ |
| Minimax | $O\left(d^3\right) + O\left(bd^2\right) + b\text{C}$ | $O\left(d^2\right) + O\left(bd\right) + b\text{M}$ |
| CSGD | $O\left(d^3\right) + O\left(bd^2\right) + b\text{C}$ | $O\left(d^2\right) + O\left(bd\right) + b\text{M}$ |

## 4.2 Relation to BSGD

Our approach is close to the BSGD approach of Chen et al. (2020). BSGD is based on the following update formula:

$$\theta_{t+1} = \theta_t - a_t \nabla l\left(\theta_t; \mathcal{S}_t\right)$$

where $\mathcal{S}_t$ are an independent random batch of indices and

$$l\left(\theta; \mathcal{S}\right) = \mathbf{y}_\mathcal{S}^T \left(K_{\mathcal{S}\mathcal{S}}\left(\theta\right) + \sigma^2 I\right)^{-1} \mathbf{y}_S + \log\left|K_{\mathcal{S}\mathcal{S}}\left(\theta\right) + \sigma^2 I\right|.$$

In our case, since $k_\alpha\left(x\right) = \phi_\alpha\left(x\right)^T \phi_\alpha\left(x\right)$, we can write $l\left(\theta; \mathcal{S}\right)$ as

$$l\left(\theta; \mathcal{S}\right) = \sum_i g_i\left(\theta\right) + h\left(\sum_i F_i\left(\theta\right)\right).$$

This leads to the update formula

$$\theta_{t+1} = \theta_t - a_t \sum_{i \in \mathcal{S}} \nabla\left[g_i\left(\theta_t\right) + \left\langle \tilde{F}_t^{-1}, F_i\left(\theta\right)\right\rangle\right], \text{ where } \tilde{F}_t = \sum_{i \in \mathcal{S}} F_i\left(\theta_t\right)$$

Writing the update formula of the BSGD algorithm this way emphasizes the fact that the only difference between the BSGD and the SCGD algorithms lays in the way in which $F\left(\theta_t\right)$ is approximated: using $\tilde{F}_t = \sum_{i \in \mathcal{S}} F_i\left(\theta_t\right)$ in BSGD, and $\tilde{F}_{t+1} = \left(1 - b_t\right)\tilde{F}_t + b_t \frac{n}{|\mathcal{S}|}\sum_{i \in \mathcal{S}} F_i\left(\theta_t\right)$ in SCGD. Intuitively, the exponential smoothing that occurs SCGD should provide additional numerical robustness beyond the theoretical advantage of converging to a stationary point without an error which depends on the batch size.

We can also see that in BSGD the scale of $\tilde{F}_t$ is incorrect because it misses the multiplication of $\sum_{i \in \mathcal{S}} F_i\left(\theta_t\right)$ by $\frac{n}{|\mathcal{S}|}$.

## 5 Experimental Result

In our experiments we consider a covariance function of the form

$$k_\alpha\left(x, x'\right) = k_u'\left(g_w\left(x\right), g_w\left(x'\right)\right),$$

where $g_w$ is a neural network comprised of two fully connected layers, both with output dimension of 128 and ReLU activation function, and $k_u'$ which is either the linear kernel

$$k_u'\left(z, z'\right) = \left\langle z, z'\right\rangle$$

or the Gaussian kernel parametrized with two hyper parameter, length scale $u_1$ and magnitude $u_2$, that is

$$k_u'\left(z, z'\right) = u_2 e^{-\frac{\|z - z'\|^2}{2u_1^2}}$$

Table 2: Negative log marginal likelihood with neural network and Linear kernel.

| batch size | method name | MINIMAX(Ours) | SCGD(Ours) | SVGP | BSGD |
|---|---|---|---|---|---|
| 32 | bike | **-1.482±0.249** | -1.454±0.264 | -0.895±0.296 | -1.167±0.701 |
| | elevators | **-1.588±0.287** | -1.530±0.195 | -0.150±0.436 | -1.125±0.017 |
| | keggdirected | **-0.745±0.076** | -0.740±0.093 | -0.151±1.057 | -0.688±0.018 |
| | keggundirected | -0.737±0.004 | **-0.739±0.011** | -0.655±0.007 | -0.716±0.025 |
| | kin40k | -1.713±0.063 | **-1.760±0.090** | 0.792±0.881 | -0.684±0.028 |
| | pol | 2.128±0.097 | 2.016±0.126 | **1.643±0.108** | 2.689±0.175 |
| | protein | **0.392±0.060** | 0.393±0.053 | 0.848±0.218 | 0.640±0.016 |
| | slice | 0.436±0.836 | 0.500±0.855 | **-0.045±0.127** | 0.459±0.089 |
| | tamielectric | **0.178±0.001** | 0.179±0.001 | 0.179±0.001 | 0.179±0.001 |
| 64 | bike | **-1.378±0.275** | -1.366±0.259 | -0.644±0.297 | -1.282±0.617 |
| | elevators | **-1.523±0.276** | -1.427±0.173 | -0.152±0.440 | -1.139±0.019 |
| | keggdirected | **-0.780±0.059** | -0.748±0.105 | -0.126±1.042 | -0.726±0.020 |
| | keggundirected | **-0.747±0.004** | **-0.747±0.005** | -0.670±0.005 | -0.710±0.015 |
| | kin40k | -1.654±0.045 | **-1.682±0.110** | 0.298±0.970 | -0.952±0.059 |
| | pol | 2.206±0.299 | 2.171±0.250 | **1.610±0.128** | 2.413±0.069 |
| | protein | **0.325±0.013** | 0.381±0.060 | 0.659±0.104 | 0.618±0.021 |
| | slice | 0.471±1.059 | 0.086±0.110 | **-0.162±0.164** | 0.304±0.143 |
| | tamielectric | **0.178±0.001** | **0.178±0.001** | 0.179±0.001 | 0.179±0.001 |
| 128 | bike | -1.340±0.039 | **-1.358±0.029** | -0.832±0.231 | -1.311±0.126 |
| | elevators | **-1.678±0.018** | -1.399±0.022 | 0.044±0.003 | -1.173±0.005 |
| | keggdirected | **-0.805±0.042** | -0.778±0.091 | -0.189±1.078 | -0.763±0.009 |
| | keggundirected | -0.725±0.022 | **-0.750±0.007** | -0.679±0.007 | -0.735±0.013 |
| | kin40k | -1.590±0.024 | **-1.618±0.030** | 0.003±0.536 | -1.224±0.038 |
| | pol | 2.215±0.156 | 2.150±0.140 | **1.583±0.014** | 2.223±0.037 |
| | protein | **0.289±0.008** | 0.327±0.010 | 0.709±0.069 | 0.576±0.003 |
| | slice | 0.019±0.043 | 0.044±0.068 | **-0.197±0.024** | -0.038±0.119 |
| | tamielectric | **0.178±0.001** | **0.178±0.001** | 0.179±0.001 | 0.179±0.001 |

Table 3: Negative log marginal likelihood with neural network and Gaussian kernel.

| batch size | method name | MINIMAX(Ours) | SCGD(Ours) | SVGP | BSGD |
|---|---|---|---|---|---|
| 16 | bike | -1.013±0.082 | -1.005±0.040 | **-1.038±0.029** | 0.105±0.107 |
| | elevators | -1.094±0.018 | **-1.135±0.009** | -0.915±0.012 | -1.081±0.014 |
| | keggdirected | **-0.755±0.004** | -0.754±0.007 | -0.640±0.012 | -0.566±0.012 |
| | keggundirected | **-0.714±0.000** | **-0.714±0.000** | -0.641±0.000 | -0.615±0.000 |
| | kin40k | -1.559±0.015 | **-1.606±0.031** | -0.697±0.027 | -0.515±0.042 |
| | pol | 2.066±0.029 | 1.980±0.027 | **1.892±0.035** | 2.656±0.192 |
| | protein | 0.485±0.007 | **0.476±0.007** | 0.625±0.016 | 0.716±0.010 |
| | slice | 1.548±0.058 | 1.556±0.042 | **0.369±0.062** | 0.461±0.143 |
| | tamielectric | **0.179±0.001** | **0.179±0.001** | **0.179±0.001** | **0.179±0.001** |
| 32 | bike | -0.929±0.023 | -0.968±0.065 | **-1.098±0.030** | 0.083±0.055 |
| | elevators | -1.042±0.023 | **-1.111±0.016** | -0.926±0.007 | -1.092±0.011 |
| | keggdirected | -0.764±0.007 | **-0.767±0.007** | -0.661±0.007 | -0.650±0.012 |
| | keggundirected | -0.735±0.000 | **-0.737±0.000** | -0.656±0.000 | -0.714±0.000 |
| | kin40k | **-1.585±0.018** | -1.571±0.017 | -0.796±0.031 | -0.695±0.021 |
| | pol | 2.115±0.027 | 2.112±0.029 | **1.807±0.023** | 2.521±0.192 |
| | protein | **0.479±0.008** | **0.479±0.004** | 0.595±0.006 | 0.674±0.009 |
| | slice | 1.950±0.120 | 1.815±0.142 | **0.159±0.042** | 0.248±0.098 |
| | tamielectric | **0.179±0.001** | **0.179±0.001** | **0.179±0.001** | **0.179±0.001** |
| 64 | bike | -0.912±0.035 | -0.946±0.049 | **-1.153±0.011** | -0.114±0.060 |
| | elevators | -0.988±0.013 | -1.056±0.018 | -0.932±0.004 | **-1.111±0.007** |
| | keggdirected | -0.762±0.006 | **-0.771±0.009** | -0.672±0.005 | -0.694±0.008 |
| | keggundirected | -0.739±0.000 | **-0.743±0.000** | -0.666±0.000 | -0.691±0.000 |
| | kin40k | **-1.547±0.022** | -1.539±0.039 | -0.864±0.026 | -0.947±0.040 |
| | pol | 2.433±0.081 | 2.399±0.059 | **1.754±0.023** | 2.237±0.030 |
| | protein | 0.479±0.007 | **0.474±0.007** | 0.585±0.011 | 0.644±0.011 |
| | slice | 1.964±0.846 | 1.899±0.764 | 0.092±0.036 | **-0.018±0.059** |
| | tamielectric | **0.179±0.001** | **0.179±0.001** | **0.179±0.001** | **0.179±0.001** |

**Setup:** The experiment is designed such that we can see the influence of the batch size on the result. We used nine regression datasets from the UCI repository (Asuncion & Newman, 2007), all with number of samples above 14,000 and less the 60,000. We tested the algorithms using different batch sizes: for the linear kernel we examined batch sizes of 32, 64, 128, 256 and 512, and for the Gaussian kernel we examined batch sizes of 16, 32, 64, 128 and 256. For SVGP, we set the number of inducing point equal to batch size. For our algorithms, labeled MINIMAX (Subsection 3.1) and SCGD (Subsection 3.2), we approximated the Gaussian kernel $k'_u$ using the *Random Fourier Features* method with random features $\varphi$ of dimension 1000 such that

$$k'_u(z, z') \approx \langle \varphi(z), \varphi(z') \rangle.$$

We ran each test on five different splits of 90% train 10% test. We used AdaDelta for all methods and for each combination of dataset, split and method, and used grid search in order to select the learning rate that achieves minimal marginal likelihood. For SCGD, we fixed $b_t = 0.9$. Note that this is very close to 1 which means that SCGD becomes quite similar to BSGD and much of the improvement comes just from the correct scaling of $\tilde{F}_t$ . For MINIMAX we fixed $\mu_t = 1.0$, since we found it enough for achieving good results despite that in theory it should be increased in an outer loop. We ran each algorithm for 100 epochs and used the hyperparameters from the iteration in which the marginal likelihood achieved its minimal value.

**Result with linear kernel - the exact case:** It seems that the fact that our algorithm does an exact stochastic optimization brings a significant improvement over existing methods in the optimization of the marginal likelihood. As expected, we can see that this advantage is more significant when the batch size is smaller (see Table 2). However, lower negative log marginal likelihood does not always translate to lower MSE on the test set as we can see in Table 4. We saw that sometimes by using a suboptimal learning rate that does not achieve the minimal loss our algorithms can achieve better results in terms of test RMSE. However, since our work is focused on optimization, we do not use procedures such as early stopping or cross validation which could potentially improve the result from the test RMSE perspective.

**Result with Gaussian kernel:** Since here for our algorithms, MINIMAX and SCGD, we use an approximated Gaussian kernel based random Fourier features, we are no longer performing an exact optimization in this case. However, we can see in Table 2 that although we use an approximated kernel eventually when the restrictions on batch size are high our methods do a better job than the existing methods in the optimization of the marginal likelihood. The advantage of our methods in the optimization is also reflected in the test error (see Table 5).

We see that in both cases, the finite dimensional RKHS and the infinite dimensional RKHS, unlike the existing inference algorithms, there is no degradation in the result of our algorithms with the decreasing of the batch size. This property can be vital for inference on weak edge devices where the memory restrictions limit the possible batch size.

## 6 Conclusion

In many cases the covariance function of a GP is defined as an inner product between features of a finite and moderate dimension. In this case, the problem of minimizing the negative-log-marginal-likelihood has a shape of a standard ridge regression problem with a non standard regularization term in the form of the log-determinant of the covariance matrix of the representations plus $\sigma^2 I$. In this work we developed two techniques that enable solving this problem with stochastic mini-batches which unlike the exiting methods does not depend on large batches in order to be exact. When the inference involves forward and backward passes of a feed forward neural net this property is of great importance and can be an enabler of such inference architectures on weak edge devices.

**Limitations:** We remark that in comparison to BSGD, both MINIMAX and SCGD are more complex and require tuning of additional hyperparameters in order to achieve the minimal negative log marginal likelihood. In addition, the optimization advantage is not fully reflected in the test error, so cross-validation is still needed to select the best algorithm.

Table 4: RMSE with neural network and Linear kernel.

| batch size | method name | MINIMAX(Ours) | SCGD(Ours) | SVGP | BSGD |
|---|---|---|---|---|---|
| 32 | bike | 0.099±0.011 | 0.095±0.007 | 0.093±0.055 | **0.082±0.073** |
| | elevators | 0.111±0.008 | 0.158±0.114 | 0.217±0.071 | **0.090±0.003** |
| | keggdirected | **0.120±0.011** | 0.123±0.015 | 0.376±0.557 | 0.122±0.009 |
| | keggundirected | 0.120±0.006 | 0.121±0.008 | **0.119±0.003** | **0.119±0.003** |
| | kin40k | 0.061±0.012 | **0.056±0.011** | 0.651±0.310 | 0.126±0.010 |
| | pol | **2.280±0.130** | 2.328±0.123 | 2.593±0.174 | 3.402±0.152 |
| | protein | **0.426±0.024** | 0.428±0.023 | 0.550±0.079 | 0.483±0.006 |
| | slice | 0.642±0.327 | 0.697±0.282 | **0.591±0.119** | 0.992±0.339 |
| | tamielectric | 0.290±0.002 | 0.289±0.002 | 0.289±0.002 | 0.289±0.002 |
| 64 | bike | 0.112±0.012 | 0.108±0.012 | 0.127±0.054 | **0.078±0.064** |
| | elevators | 0.111±0.013 | 0.114±0.023 | 0.216±0.071 | **0.094±0.010** |
| | keggdirected | **0.118±0.008** | 0.120±0.011 | 0.377±0.557 | 0.119±0.009 |
| | keggundirected | **0.119±0.006** | 0.121±0.008 | 0.121±0.008 | 0.123±0.006 |
| | kin40k | 0.065±0.015 | **0.060±0.010** | 0.443±0.338 | 0.097±0.009 |
| | pol | 2.449±0.160 | **2.263±0.151** | 2.415±0.115 | 3.067±0.145 |
| | protein | 0.430±0.007 | **0.426±0.021** | 0.494±0.046 | 0.473±0.003 |
| | slice | **0.479±0.066** | 0.615±0.288 | 0.554±0.160 | 0.680±0.173 |
| | tamielectric | 0.290±0.002 | 0.289±0.002 | 0.289±0.002 | 0.289±0.002 |
| 128 | bike | 0.125±0.005 | 0.123±0.011 | 0.111±0.024 | **0.056±0.004** |
| | elevators | 0.110±0.004 | 0.101±0.002 | 0.250±0.007 | **0.091±0.003** |
| | keggdirected | **0.116±0.008** | 0.118±0.011 | 0.370±0.561 | 0.123±0.018 |
| | keggundirected | 0.124±0.008 | 0.121±0.007 | **0.118±0.006** | 0.119±0.007 |
| | kin40k | 0.073±0.002 | **0.071±0.003** | 0.283±0.193 | 0.078±0.003 |
| | pol | 2.469±0.174 | **2.413±0.234** | 2.537±0.094 | 2.642±0.110 |
| | protein | **0.435±0.009** | 0.436±0.009 | 0.513±0.023 | 0.461±0.006 |
| | slice | **0.528±0.086** | 0.546±0.181 | 0.542±0.111 | 0.583±0.086 |
| | tamielectric | 0.290±0.002 | 0.289±0.002 | 0.289±0.002 | 0.289±0.002 |

Table 5: RMSE with neural network and Gaussian kernel.

| batch size | method name | MINIMAX(Ours) | SCGD(Ours) | SVGP | BSGD |
|---|---|---|---|---|---|
| 16 | bike | 0.081±0.012 | 0.096±0.022 | **0.046±0.007** | 0.220±0.014 |
| | elevators | 0.096±0.003 | 0.098±0.004 | **0.089±0.002** | **0.089±0.003** |
| | keggdirected | **0.118±0.008** | **0.118±0.007** | 0.122±0.005 | 0.127±0.005 |
| | keggundirected | 0.131±0.000 | 0.128±0.000 | **0.116±0.000** | 0.138±0.000 |
| | kin40k | **0.039±0.002** | 0.040±0.002 | 0.110±0.007 | 0.147±0.007 |
| | pol | 2.212±0.146 | **2.200±0.104** | 2.674±0.081 | 3.774±0.440 |
| | protein | **0.387±0.009** | 0.388±0.009 | 0.471±0.008 | 0.507±0.012 |
| | slice | 0.519±0.149 | **0.496±0.110** | 0.562±0.067 | 0.661±0.090 |
| | tamielectric | **0.289±0.002** | **0.289±0.002** | **0.289±0.002** | **0.289±0.002** |
| 32 | bike | 0.093±0.010 | 0.095±0.012 | **0.047±0.010** | 0.235±0.024 |
| | elevators | 0.100±0.004 | 0.098±0.004 | **0.088±0.003** | 0.089±0.002 |
| | keggdirected | **0.118±0.008** | **0.118±0.007** | 0.121±0.007 | 0.125±0.008 |
| | keggundirected | 0.130±0.000 | 0.128±0.000 | **0.115±0.000** | 0.116±0.000 |
| | kin40k | **0.039±0.002** | **0.039±0.001** | 0.096±0.005 | 0.124±0.003 |
| | pol | 2.210±0.133 | **2.209±0.080** | 2.552±0.117 | 3.338±0.151 |
| | protein | **0.387±0.008** | 0.389±0.010 | 0.461±0.009 | 0.493±0.007 |
| | slice | **0.511±0.039** | 0.515±0.065 | 0.527±0.139 | 0.565±0.102 |
| | tamielectric | **0.289±0.002** | **0.289±0.002** | **0.289±0.002** | **0.289±0.001** |
| 64 | bike | 0.101±0.008 | 0.106±0.014 | **0.035±0.001** | 0.209±0.009 |
| | elevators | 0.089±0.003 | 0.098±0.007 | **0.088±0.003** | **0.088±0.003** |
| | keggdirected | 0.118±0.008 | **0.117±0.008** | 0.119±0.006 | 0.122±0.007 |
| | keggundirected | 0.128±0.000 | 0.127±0.000 | **0.113±0.000** | 0.117±0.000 |
| | kin40k | **0.041±0.001** | **0.041±0.002** | 0.088±0.005 | 0.096±0.003 |
| | pol | **2.348±0.077** | 2.353±0.088 | 2.512±0.081 | 3.020±0.071 |
| | protein | **0.388±0.008** | 0.389±0.005 | 0.456±0.005 | 0.481±0.004 |
| | slice | 0.616±0.163 | 0.546±0.129 | **0.507±0.116** | 0.512±0.120 |
| | tamielectric | **0.289±0.002** | **0.289±0.002** | **0.289±0.002** | **0.289±0.002** |

**Limitations:** However, we remark that in comparison to BSGD, both MINIMAX and SCGD are more complex and require tuning of additional hyperparameters in order to achieve the minimal negative log marginal likelihood. In addition, the optimization advantage is not fully reflected in the test error, so cross-validation is still needed to select the best algorithm.

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

# A  Complexity Analysis

We analyze the computational complexity for each one of the algorithms. We ignore the complexity of for the computation of $\phi_\alpha(x_i)$ and $\frac{\phi_\alpha(x_i)}{\partial \alpha}$ as they are the same for all the algorithms. We let $d$ be the dimension of $\phi_\alpha(x_i)$ and $b$ be the batch size.

## A.1  Minimax

**Computational Complexity Forward**

| Component | Complexity |
|---|---|
| $g_i$ | $O(d)$ |
| $F_i$ | $O(d^2)$ |
| $h(A)$ | $O(d^3)$ |
| $\frac{\left\langle B, \frac{1}{n}A - F_i(\theta) \right\rangle}{\|A\|}$ | $O(d^2)$ |
| **Total** | $O(d^3) + O(bd^2)$ |

**Computational Complexity Backward**

| Component | Complexity |
|---|---|
| $\frac{\partial g_i}{\partial w}$ | $O(d)$ |
| $\frac{\partial g_i}{\partial \phi_i}$ | $O(d)$ |
| $\frac{\partial h}{A} = A^{-1}$ | $O(d^3)$ |
| $\frac{\partial}{\partial B} \frac{\left\langle B, \frac{1}{n}A \right\rangle}{\|A\|} = \frac{\frac{1}{n}A}{\|A\|}$ | $O(d^2)$ |
| $\frac{\partial}{\partial A} \frac{\left\langle B, \frac{1}{n}A \right\rangle}{\|A\|} = \frac{\frac{1}{n}B\|A\| - \left\langle B, \frac{1}{n}A \right\rangle \frac{A}{\|A\|}}{\|A\|^2}$ | $O(d^2)$ |
| $\frac{\partial}{\partial \phi_i} \frac{\left\langle B, F_i \right\rangle}{\|A\|} = \frac{2B\phi_i}{\|A\|}$ | $O(d^2)$ |
| **Total** | $O(d^3) + O(bd^2)$ |

**Memory Complexity Forward**

| Component | Complexity |
|---|---|
| $g_i$ | $O(d)$ |
| $A, B, F_i$ | $O(d^2)$ |
| **Total** | $O(d^2)$ |

**Memory Complexity Backward**

| Component | Complexity |
|---|---|
| $\frac{\partial g_i}{\partial w}$ | $O(d)$ |
| $\frac{\partial g_i}{\partial \phi_i}$ | $O(d)$ |
| $\frac{\partial h}{A} = A^{-1}$ | $O(d^2)$ |
| $\frac{\partial}{\partial B} \frac{\left\langle B, \frac{1}{n}A \right\rangle}{\|A\|} = \frac{\frac{1}{n}A}{\|A\|}$ | $O(d^2)$ |
| $\frac{\partial}{\partial A} \frac{\left\langle B, \frac{1}{n}A \right\rangle}{\|A\|} = \frac{\frac{1}{n}B\|A\| - \left\langle B, \frac{1}{n}A \right\rangle \frac{A}{\|A\|}}{\|A\|^2}$ | $O(d^2)$ |
| $\frac{\partial}{\partial \phi_i} \frac{\left\langle B, F_i \right\rangle}{\|A\|} = \frac{2B\phi_i}{\|A\|}$ | $O(d)$ |
| **Total** | $O(d^2) + O(bd)$ |

## A.2 SCGD

**Computational Complexity Forward**

| Component | Complexity |
|-----------|-----------|
| $g_i$ | $O(d)$ |
| $\tilde{F}^{-1}$ | $O(d^3)$ |
| $F_i$ | $O(d^2)$ |
| **Total** | $O(d^3)$ |

**Computational Complexity Backward**

| Component | Complexity |
|-----------|-----------|
| $\frac{\partial g_i}{\partial w}$ | $O(d)$ |
| $\frac{\partial g_i}{\partial \phi_i}$ | $O(d)$ |
| $\frac{\partial}{\partial \phi_i}\left\langle \tilde{F}_t^{-1}, F_i \right\rangle = 2\phi_i \tilde{F}_t^{-1}$ | $O(d^2)$ |
| **Total** | $O(bd^2)$ |

**Memory Complexity Forward**

| Component | Complexity |
|-----------|-----------|
| $g_i$ | $O(d)$ |
| $F_i$ | $O(d^2)$ |
| $\tilde{F}$ | $O(d^2)$ |
| **Total** | $O(d^2)$ |

**Memory Complexity Backward**

| Component | Complexity |
|-----------|-----------|
| $\frac{\partial g_i}{\partial w}$ | $O(d)$ |
| $\frac{\partial g_i}{\partial \phi_i}$ | $O(d)$ |
| $\frac{\partial}{\partial \phi_i}\left\langle \tilde{F}_t^{-1}, F_i \right\rangle = 2\phi_i \tilde{F}_t^{-1}$ | $O(d)$ |
| **Total** | $O(bd)$ |

## A.3 BSGD

$$\frac{\partial}{\partial Z}\left[\mathbf{y}^T \left(K + \sigma^2 I\right)^{-1} \mathbf{y} + \log \left|K + \sigma^2 I\right|\right]$$

$$= -2\left(K + \sigma^2 I\right)^{-1} \mathbf{y}\mathbf{y}^T \left(K + \sigma^2 I\right)^{-1} Z$$

$$+ 2\left(K + \sigma^2 I\right)^{-1} Z$$

Computational complexity: $O(b^3) + O(b^2 d)$.

Memory complexity: $O(b^2) + O(bd)$

## B   Missing Proofs

**Theorem 1.** *For all $\lambda > 0$, $V \in \mathbb{R}^{n \times d}$ and $\mathbf{b} \in \mathbb{R}^d$ we have that $\mathbf{b}^T \left(VV^T + \lambda I\right)^{-1} \mathbf{b} = \min_{\mathbf{w}} \frac{1}{\lambda} \|V\mathbf{w} - \mathbf{b}\| + \|\mathbf{w}\|^2$*

*Proof.* The minimizer of $\frac{1}{\lambda} \|V\mathbf{w} - \mathbf{b}\|^2 + \|\mathbf{w}\|^2$ is given by $\hat{\mathbf{w}} = \left(V^T V + \lambda I\right)^{-1} V^T \mathbf{b}$. With this we can calculate

$$
\begin{aligned}
\min_{\mathbf{w}} \frac{1}{\lambda} \|V\mathbf{w} - \mathbf{b}\|^2 + \|\mathbf{w}\|^2 &= \frac{1}{\lambda} \|V\hat{\mathbf{w}} - \mathbf{b}\|^2 + \|\hat{\mathbf{w}}\|^2 \\
&= \frac{1}{\lambda} \left[ \hat{\mathbf{w}}^T V^T V \hat{\mathbf{w}} - 2\mathbf{b}^T V \hat{\mathbf{w}} + \mathbf{y}^T \mathbf{y} + \lambda \hat{\mathbf{w}}^T \hat{\mathbf{w}} \right] \\
&= \frac{1}{\lambda} \left[ \mathbf{b}^T \mathbf{b} - \left( 2\mathbf{b}^T V \hat{\mathbf{w}} - \hat{\mathbf{w}}^T \left(V^T V + \lambda I\right) \hat{\mathbf{w}} \right) \right] \\
&= \frac{1}{\lambda} \Big[ \mathbf{b}^T \mathbf{b} - \Big( 2\mathbf{b}^T V \left(V^T V + \lambda I\right)^{-1} V^T \mathbf{b} \\
&\quad - \mathbf{b}^T V \left(V^T V + \lambda I\right)^{-1} \left(V^T V + I\right) \left(V^T V + \lambda I\right)^{-1} V^T \Big) \mathbf{b} \Big] \\
&= \frac{1}{\lambda} \mathbf{b}^T \left[ I - V \left(V^T V + \lambda I\right)^{-1} V^T \right] \mathbf{b} \\
&= \mathbf{b}^T \left(VV^T + \lambda I\right)^{-1} \mathbf{b} \quad \text{(by Woodbury formula)}
\end{aligned}
$$

$\square$

