# OpenReview forum: "Unbiased Stochastic Optimization for Gaussian Processes on Finite Dimensional RKHS"
_TMLR — Rejected by TMLR_

### Review · Reviewer_8gDy · 2025-09-22

**Summary Of Contributions:**

**SUMMARY**: The paper proposes two mini-batch stochastic algorithms for hyperparameter learning in Gaussian Process Regression when the kernel admits a finite (moderate)–dimensional feature map: (i) a minimax reformulation that replaces the non-decomposable log-determinant with a constrained saddle problem, and (ii) an application of Stochastic Compositional Gradient Descent (SCGD) using an exponential-average estimate. The authors claim these methods give exact stochastic updates in the finite-dimensional RKHS case and empirically outperform Biased SGD (BSGD) and SVGP when batch size is constrained. The paper evaluates the algorithms on nine UCI regression datasets using primarily the negative log marginal likelihood which shows that MINIMAX or SCGD generally finds solutions with lower negative marginal log likelihood than standard SGD methods and SVI alternatives.



**OVERALL**: the paper addresses an important and practical issue — stochastic hyperparameter optimization of GPs when memory limits force small batches. The focused setting (finite-dimensional RKHS, and the RFF extension for infinite case) makes the problem concrete. The proposed algorithms are reasonable and well-motivated by prior literature. It is a niche topic but certainly of relevance to a subpart of the TMLR audience. However, there are missing points and various aspects especially related to the experimental setup that need to be improved before I can recommend acceptance.

**Additional Comments:**

*MINOR COMMENTS/SUGGESTIONS:*

- Throughout: Inconsistent use of punctuation after equations. I ***strongly suggest*** to fix this.
- Abstract/throughout: Please clarify what "moderate" dimension means quantitatively.
- P1 Motivation: “ …approximations are required…” -> There are methods scaling exact GPs to “big data” in certain circumstances (e.g. https://arxiv.org/pdf/1903.08114). I ***suggest*** mentioning this and state why the work is still relevant as hyperparameters are still in need of estimation.
- P2 “…based on computing stochastic batches….” -> I find the formulation unclear - what does “computing stochastic batches” mean exactly?
- P2, mid: “… additional unnecessary approximation…” -> Not clear to me that it is an “unnecessary” approximation – could you elaborate please?
- P5 “log det F(\theta)” -> log |F(\theta )|.
- P5: It would be pertinent to formally define $\epsilon$ (see also comment about formal theorems). ***Strongly suggest*** to fix this.
- P5: $\sigma_min$ -> $\sigma_{min}$
- P6: ”AD” is undefined. I ***strongly suggest*** to fix this.
- P10: AdaDelta -> I appreciate it is a standard algorithm, but I ***strongly suggest*** to include reference for completeness.
- P10: I find a few sentences hard to pass, e.g. “We saw that sometimes….”, “It seems that the fact that our algorithm does …”, “We see that in both cases….”. ***Strongly suggest*** to fix this.

**Audience:**

Yes

**Audience Explanation:**

The sub-community interested in Gaussian Processes and other fields doing stochastic gradient decent in correlated settings would be interested in the empirical insights provided in the paper.

**Broader Impact Concerns:**

No concerns.

**Claims And Evidence:**

No

**Claims Explanation:**

**MAIN COMMENTS/QUESTIONS/SUGGESTIONS**:

-	**Context and presentation of problem**: The GPR setup and context is only just clear enough, and I think readers would feel more supported if the paper included a bit more context/background (sec. 2.2). I'd ***suggest*** starting from a Bayesian regression model with Normal likelihood and GP prior leading to a closed form predictive distribution but conditioned on hyperparameters. This leads to either a full Bayesian problem of estimating hyperparameters or integrating them out; however, the problem is typically reduced to a ML-II type inference which is considered here.
The underlying problem of optimizing the marginal likelihood is very briefly stated on p. 4 (top). I'd ***suggest*** guiding the reader and spell out the core problem to put emphasis on the key problem to be tackled and the importance of doing so.


-	**Presentation of the methods (sec 3)**: The presentation of the two methods is very concise. I’d ***suggest*** the authors ensure somewhat better flow by adding a bit more boilerplate text to ensure smooth and natural flow in the section (e.g. going from 3.1 to 3.2). On a similar note, perhaps consider if sec 4.2 could be integrated fluently in sec 3.


-	**Missing/insufficient formal convergence statements and assumptions**: The paper appeals to existing theory (Boţ & Böhm 2020 for minimax and Wang et al. 2017 for SCGD) but does not provide a **self-contained** theorem or similar that states under what precise assumptions the proposed instantiations converge to stationary points of the original objective l(\theta). I’d ***strongly suggest*** adding this ideally along with clear proof sketches arguing how the cited references apply (potentially in the appendix).


-	**Mismatch between theoretical procedure and experimental setup**: The minimax formulation in Sec. 3.1 relies on taking penalty parameter $\mu_k \to \infty$ to recover the constrained problem, yet in experiments the authors fix $\mu_t=1.0$ (Sec. 5). Similarly, SCGD theory requires specific decreasing schedules $a_t$,$b_t$, but experiments use constants.  It is argued that in practice this does not matter; however, for a paper that proposes two new algorithms for a specific problem, I’d ***strongly suggest*** to evaluate the full/correct algorithm initially and contrast it to any simplified versions. As is, claims of "exactness" are not supported by the practical setup.

-	**Algorithm presentations**: … in connection with the above, please ***consider*** including algorithm boxes to clearly spell out the steps involved (perhaps in the appendix).


-	**SVI baseline - number of inducing points**: I am not sure I fully understand or agree with the arguments behind the number of inducing points in the SVI baseline.
  - Firstly, sec 4.1 states that number is “in the same order as b” and “roughly the same as the mini-batch size” which is quite vague. I'd ***strongly suggest*** the authors clarify this and explain exactly how the number is determined and what the values are for each of the UCI datasets? E.g. is it based on matching the actual peak memory consumption of the methods?
  - Secondly, I’d generally expect the number of inducing points to depend on the complexity of the problem (i.e. an actual hyperparameter). I’d ***strongly suggest*** providing an in-depth experiment with the SVI baseline (for at least a subset of problems) where induction points are empirically optimized (e.g. via grid search) and the actual memory and compute is monitored to shed some light on this aspect and ensure a fair comparison.


-	**Numerical insights, convergence and exactness**: Given paper’s focus, I believe the numerical properties of the method should be demonstrated more clearly to convince the reader that it actually finds something useful not only reporting its loss value (potentially at a stationary point) but also show how the optimization converges to an unbiased estimate of the actual parameters. Therefore, I’d ***strongly recommend*** including (at least) one experiment on a synthetic dataset e.g. sampled from fixed models with known hyperparameters showing how the parameter estimates converge etc. See Chen (2020) Figure 1 and 2 for inspiration.


-	**Metrics**: I’d ***strongly suggest*** including (negative) predictive log likelihood in combination with (or instead of) RMSE to capture the predictive uncertainty.


-	**Scalability and complexity**: The complexity analysis is informative but when dealing with concrete datasets, it is often helpful and more convincing to also report the wall clock time and (peak) memory consumption. I'd therefore ***strongly suggest*** adding this information to provide a more convincing argument related to the practical application of their methods compared to alternatives.


-	**Effect of batch size**: It is difficult to appreciate the details and patterns in table 1-4 given the many conditions and numbers. To supplement the table and provide a more convincing story to back up the claims, I’d ***strongly suggest*** creating a simple figure (e.g. a line plot with up to nine graphs) showing the effect of batch size – and ideally consider a larger range of batch size than included currently.


-	**Statistical significance**: It is unclear what the number after “+/-“ represents (std. deviation, std. error etc.)…? I’d ***strongly recommend** explicitly stating in the caption what the number refers to and consider the information in the discussion of the results. Ideally, ***consider*** including formal statistical tests when referring to significance.


-	**Reproducibility**: The paper does not contain any details about implementation or training or training platform (e.g. hardware, software frameworks etc.) and whether they plan to release source code. I'd ***strongly suggest*** proving this as it would greatly improve reproducibility.

**Requested Changes:**

Apologies, I wrote my comments and suggest in one go above, but I have indicated my requested changes in the comments above (and below for the minor comments) and they should be read as follows:


 - ***strongly suggest*** / ***strongly recommend***: This is critical to ensuring my recommendation unless strong arguments are provided against the recommendation.


- ***consider*** / ***suggest***: These are things I feel would improve the paper but not strictly needed to ensure my recommendation.

---

### Review · Reviewer_WJ5u · 2025-09-29

**Summary Of Contributions:**

This paper introduces two new approaches for learning hyperparameters of Gaussian processes with kernels defined by a neural network. As opposed to existing methods which are typically biased when using minibatching, the proposed method is guaranteed to converge to a stationary point of the objective. Contrary to SVGP and BSGD, the proposed approaches Minimax and CSGD scale cubically in time and quadratically in space with the number of features rather than the batch size. The authors present experimental results for UCI regression datasets and compare to both SVGP and BSGD.

**Additional Comments:**

- How come you are using Adadelta for training, rather than something more standard nowadays like SGD with momentum or Adam?

**Audience:**

Yes

**Audience Explanation:**

The topic of optimizing the hyperparameters of a GP with a deep kernel is of interest to the community and scalable methods for this setting are definitely in need. It is also interesting to see the direct theoretical comparison of CSGD and BSGD which reveals the lack of batch-dependent scaling of the damped kernel matrix term ($F(\theta)$). However, the setting in which the proposed methods are useful is less clear, given that there are other methods which in experiment demonstrate scaling to larger datasets and bigger neural networks, making it hard to draw any conclusions from the current set of experiments.

**Claims And Evidence:**

No

**Claims Explanation:**

Unfortunately, in its current form the paper does not give convincing and clear experimental evidence for the need of the proposed method. Further the presentation of the work is in parts severely lacking.

**Missing comparison to simple baselines.**
The authors do not compare against a straightforward baseline that exploits the matrix inversion/determinant lemma $b=n > d$. For most of the datasets considered, an $O(n^2 d)$ computation is feasible and would provide a clear picture of the peak performance achievable by exact marginal log-likelihood. Without this, it is difficult to assess the value of the proposed approximation.

Also, the experiments omit a direct comparison to KISS-GP, the baseline from the original deep kernel learning paper (Wilson et al. 2016). That work demonstrates scalability to datasets with up to 2M points and uses a substantially larger fully connected neural network (architecture: d_in=1000-1000-500-50-2) than the one tested here. Since KISS-GP is also linear in the number of training points, it is unclear from the current results why the proposed approach is necessary or preferable.

**Batch size choices.**
The batch sizes used in the experiments are very small given that only a two-hidden-layer ReLU network with width=128 is used. In practice, much larger batches are possible in this setting—even on edge devices. Thus, the experiments do not convincingly evaluate the regime that the paper claims to target. To make a convincing case, either larger networks or larger batch sizes should be used for the experiments.

**Scalability concerns.**
Both MINIMAX and CSGD scale cubically in the number of network parameters, which quickly becomes prohibitive. This limits the method’s usefulness when expressivity requires a deep kernel. This scaling issue may explain why the neural network used here (~33k parameters) is far smaller than in prior work (e.g., ~1.5M parameters in the DKL paper). The paper should explicitly discuss this limitation.

**Lack of runtime comparisons.**
The paper reports results under a fixed epoch budget but does not include wall-clock runtime experiments. Without these, it is unclear whether the proposed method is competitive in practice. On modern hardware complexity analysis alone can be misleading, especially when parallelizing computations on a GPU.

**Presentation of the work.**
Unfortunately, in addition to the concerns expressed above, the submission in its current form did clearly not receive the attention a TMLR paper should have. The results section mentions non-existing results for batch sizes 256 and 512 overstating the actual experiments. Further, the results for the Gaussian kernel reference the wrong table and the limitation section is duplicated in the paper. While the methods proposed in this work may very well be interesting and worthy of publication, the lack of care in presenting the results and therefore the lack of respect for the reviewing process is at the very least frustrating.

**Requested Changes:**

### Critical Changes
- Report the missing experimental results for batch sizes 256 and 512 as stated in Section 5.
- Show learning curves (train and test error) for the most representative datasets for all methods both as a function of epochs and wallclock time, not just the best test error across epochs in a table. This provides a more complete comparison between methods, for example speed of convergence, and also shows any runtime differences.
- Plot the final test error as a function of batch size to see where the point is at which a large enough batch size results in SVGP / BSGD performing the same as your approaches. This is useful to understand what regime your method is most useful in. This is also important to back up the claim that your method does not degrade with batch size, which is not easy to discern from the table.
- Delete one of the duplicated limitations sections.

### Other Suggested Improvements
- Please include information about the UCI datasets (size, input dimension) and the appropriate individual citations for the datasets.
- Submission of code in the supplementary material would strengthen the submission.
- Section 2.2: "$y$ is a sample of the random vector $f(X) + \sigma^2 \epsilon$ should be $\sigma$ not $\sigma^2$
- Results with Gaussian kernel: References wrong table for NLL results, should be Table 3

---

### Review · Reviewer_vTJv · 2025-10-10

**Summary Of Contributions:**

This paper addresses the problem of applying stochastic gradient descent to the problem of hyperparameter learning in the context of Gaussian processes. The most innovative contribution in my opinion is the reframing of the maximization problem as a minimax problem via a parameter expansion. This reframing permits unbiased estimation of gradients based on mini-batches, and the application of recent work in constrained optimization.

Additional contributions from this paper include the application of Stochastic Compositional Gradient Descent to the GP hyperparameter problem. This idea is also neat and relatively simple to implement, thus despite being less innovative, I think it is probably more practical.

Overall, this paper presents an interesting idea from a theoretical perspective which itself will be useful to other researchers. The main weakness is in the experiments section, which shows only slight improvement over existing methods, and could be fleshed out more completely.

**Audience:**

Yes

**Audience Explanation:**

Hyperparameter optimization is a critical step of any GP workflow, often being repeated many times such as in Bayesian optimization. It is not unreasonable to assume that in large data or small memory scenarios, a full pass of the data may be a nuisance. In fact, I have seen in my own work instances when stochastic methods might have been of use, thus any improvement in stochastic methods for hyper parameter optimization is welcome.

**Broader Impact Concerns:**

There are no broader impact concerns.

**Claims And Evidence:**

Yes

**Claims Explanation:**

Experimental results are interpreted accurately and math is sufficiently rigorous.

**Requested Changes:**

- More discussion of how the projections are actually performed on section 3.1., and the associated computational and memory costs (would strengthen)
- Several times it is mentioned that previous theoretical results apply under relatively mild conditions. Have the authors actually verified that their problem formulation actually satisfies these conditions? Even if they are mild it would be good to note (would strengthen)
- Computational and memory complexity are discussed in theory, but not reported in the experiments section. It should be relatively easy to compare runtimes between the different algorithms (would strengthen)

---

### Decision · Action_Editor_h8SE · 2025-11-30

**Recommendation:** Reject

**Audience:**

Yes

**Audience Explanation:**

Scalable Gaussian process hyperparameter learning is a topic of interest to the TMLR community. While the focus of this paper, degenerate Gaussian processes corresponding to finite-dimensional function spaces, is more niche, it still meets the requirements for TMLR.

**Claims And Evidence:**

No

**Claims Explanation:**

While the mathematics behind the method are generally sound, there are several issues with the experiments and presentation. The most notable issues are:

- Missing obvious baselines, such as direct solves with the matrix inversion lemma
- Missing wall-clock times, which would justify the scalability
- Lack of discussion or analysis around the number of network parameters
- Lack of a formal convergence statement with assumptions
- Missing experiments with 256 and 512 batch size, despite being promised in the manuscript
- Experiments on synthetic datasets to empirically demonstrate recovering the true hyperparameters with the proposed method.

As it stands, these numerous issues leave the claims only partially supported. The authors did not respond to these concerns, despite being given an extension during the response period.

**Resubmission Of Major Revision:**

The authors may consider submitting a major revision at a later time.